# Nursing Interventions for Client and Family Training in the Proper Use of Noninvasive Ventilation in the Transition from Hospital to Community: A Scoping Review

**DOI:** 10.3390/healthcare12050545

**Published:** 2024-02-25

**Authors:** Jéssica Moura Gabirro Fernando, Margarida Maria Gaio Marçal, Óscar Ramos Ferreira, Cleoneide Oliveira, Larissa Pedreira, Cristina Lavareda Baixinho

**Affiliations:** 1Hospital Vila Franca de Xira, 2600-009 Vila Franca de Xira, Portugal; jessicafernando@esel.pt; 2Department of Fundamentals of Nursing, Escola Superior de Enfermagem de Lisboa, Nursing School of Lisbon, 1600-190 Lisbon, Portugal; margaridamarcal@campus.esel.pt (M.M.G.M.); oferreira@esel.pt (Ó.R.F.); 3Nursing Research, Innovation and Development Centre of Lisbon (CIDNUR), 1900-160 Lisbon, Portugal; 4Medical School Estácio Idomed Quixadá, University Center Estacio do Cearà, Fortaleza 60035-111, Brazil; cleo_sbf@yahoo.com.br; 5Nursing School, Federal University of Bahia, Salvador 40170-110, Brazil; larissa.pedreira@uol.com.br; 6Center of Innovative Care and Health Technology (ciTechCare), 2414-016 Leiria, Portugal

**Keywords:** noninvasive ventilation, transitional care, training, return home, discharge

## Abstract

Noninvasive ventilation is an increasingly disseminated therapeutic option, which is explained by increases in the prevalence of chronic respiratory diseases, life expectancy, and the effectiveness of this type of respiratory support. This literature review observes that upon returning home after hospital discharge, there are difficulties in adhering to and maintaining this therapy. The aim of this study is to identify nursing interventions for client and family training in the proper use of noninvasive ventilation in the transition from hospital to community. A scoping review was carried out by searching MEDLINE, CINAHL, Scopus, and Web of Science. The articles were selected by two independent reviewers by applying the predefined eligibility criteria. Regarding transitional care, the authors opted to include studies about interventions to train clients and families during hospital stay, hospital discharge, transition from hospital to home, and the first 30 days after returning home. The eight included publications allowed for identification of interventions related to masks or interfaces, prevention of complications associated with noninvasive ventilation, leakage control, maintenance and cleaning of ventilators and accessories, respiratory training, ventilator monitoring, communication, and behavioral strategies as transitional care priority interventions to guarantee proper training in the transition from hospital to community.

## 1. Introduction

According to the Forum of International Respiratory Societies, respiratory diseases are one of the main causes of death and disability worldwide. One of the reasons for this is that 200 million people have chronic obstructive pulmonary disease (COPD), of whom approximately 3.2 million die every year. This problem is the third most frequent cause of death around the world. Additionally, asthma, one of the main chronic diseases, affects roughly 350 million people on the planet [1].

A research study involving 14,127 individuals ≥40 years of age with COPD observed that the 5-year mortality of COPD patients was 25.4% (29.9% in males and 19.1% in females). The mortality rate increased rapidly with age [2]. Furthermore, there are difficulties in the adherence to noninvasive ventilation (NIV) [3,4,5].

According to a 2020 report by the National Observatory of Respiratory Diseases, one of the activities offered by organizations for patients with respiratory disorders is related to promoting knowledge, information, and literacy in the different pathologies that are part of this category [2]. Consequently, and more specifically, it is crucial to educate and train clients to use NIV since there are individuals that need to continue its use after hospital discharge in order to promote care continuity, maintain their adherence to the treatment, and prevent worsening of the disease [1,3,5]. Not accomplishing these tasks implies the risk of going back to hospital. 

The probability of developing several complications associated with NIV increases proportionally with its duration, client restlessness, and a constant need to adjust the mask [6]. Therefore, preventing complications by aiming for successful use of NIV is considered one of the main functions of nurses in the care of clients in critical situations who require NIV [6].

Complications related to the use of interfaces in NIV are the most frequent. They include discomfort caused by poor adjustment of masks and the pressure exerted by them (30% to 50% of cases); the feeling of claustrophobia (5% to 10%), which can trigger client restlessness; and pressure injuries, mostly in the nasal pyramid (5% to 10%). There are also complications related to pressure and air flow: dryness of oral and nasal mucosae (20% to 50%), nasal congestion (10% to 20%), and eye irritation resulting from leakage around the mask. Additionally, “gastric distention can also affect some clients, but it is rarely intolerable”. Only 5% of clients showed more serious complications, such as aspiration pneumonia, hypotension, and pneumothorax [4].

Other causes for difficulties in adhering to the NIV are associated with non-acceptance of the need for domiciliary NIV, the fear of becoming “technology-dependent”, and psychological issues such as anxiety and claustrophobia [5]. 

The initiation and maintenance of long-term home NIV represents a significant lifestyle change which intrinsically requires sustained effort on the part of the patient and often their family or caregivers [5], who usually mention difficulty in using NIV because of a lack of information that should be provided at hospital discharge [7]. 

The studies reinforce that patients and their caregivers show gaps in knowledge and insufficient skills to maintain NIV after discharge [8,9,10], which will affect the adherence and the capacity of the caregiver to provide care, with consequences such as insufficient functional improvement and unnecessary hospital readmissions [9,10].

These problems point out the need for a special intervention by health professionals, mainly nurses, oriented toward patients and family caregivers, not just because of the transition they are about to experience into the new role they will take on, but also to allow the possibility of obtaining better health results [10].

In order to minimize these issues, it is important to focus on coordination between hospitals and primary healthcare units, which should carry out prior assessments of the homes of the most critical patients before hospital discharge, including the possibility of evaluating the implementation of ventilation therapy [2].

The literature review performed by the authors observed that there are few studies on preparing people with NIV to return home and the strategies used to empower patients and families to return home. Nurses are the professionals who have responsibility for developing interventions in the area of education for the health of people under ventilation therapy that allow proper and safe transition from hospital to home. These procedures will help these clients and their relatives or other meaningful people to develop new cognitive and instrumental competencies that can effectively guarantee care continuity at home [8].

The objective of the present study was to identify nursing interventions for client and family training in the proper use of noninvasive ventilation in the transition from hospital to community.

## 2. Materials and Methods

### 2.1. Study Design

The method chosen was a scoping review. It allows for identification of the evidence types available on the subject, examination of study types, and detection and analysis of knowledge gaps [11]. An initial search identified several studies about nursing interventions oriented toward inpatients under NIV but found that studies addressing preparation to return home and transitional care of these patients are scarce. 

The research question for the present review was formulated by applying the PCC (Population, Concept, Context) mnemonic [11]: What nursing interventions are implemented to train adult or elderly clients and family caregivers to use NIV at home after hospital discharge? 

### 2.2. Eligibility Criteria 

The object of study and the research question helped define the eligibility criteria of the studies to be included in this scoping review. To narrow the research strategy and increase the rigor and quality of the results, inclusion and exclusion criteria were defined for each of the elements of the PCC acronym, according to the methodology proposed by JBI [11] (Table 1).

Regarding transitional care, the authors opted to include studies about interventions to train clients and family caregivers during hospital stay, hospital discharge, transition from hospital to home, and the first 30 days after returning home. This option is based on the concept of transitional care, which includes the notion that interventions to manage the transition from hospital to home can occur in three distinct steps: before clients leave the hospital, at hospital discharge, and between 48 h and 30 days after hospital discharge [10,12]. To be included in the sample, the articles had to address interventions in at least one of these phases, and their results had to allow for the identification of interventions that allow for training (client and family caregiver) on the use of noninvasive ventilation in the transition from hospital to community.

The chosen publication date period was from 2017 to 2023. This time frame is justified as we sought to obtain recent studies and considering that concern about transitional care is a recent development in healthcare [12]. A full text filter was applied, and only publications written in English or Portuguese were included. There was no restriction on the country of origin of the articles, only on the language used.

### 2.3. Data Collection

The search for publications was carried out in MEDLINE (via the PubMed platform), CINAHL, Scopus, and Web of Science databases (databases available at the university) by using terms in natural language and descriptors used in the indexing of each database. Table 2 shows the terms used in the search in the MEDLINE database. 

Searches in the databases occurred in May 2023. Gray literature was also considered in the search, which included consulting Google Scholar and the Portuguese Open Access Scientific Repository. E-books, online handbooks, books, reports, dissertations, conference proceedings, documents from the ministry of health, general directorates of health, and available guidelines were accepted in this phase.

In order to facilitate article organization and selection after the search process, the Rayyan^®^ platform was used. Data screening and extraction was carried out independently by two researchers (JF and MM). When these researchers did not reach a consensus, a third researcher (CLB) was recruited to make the decision. 

Documents obtained from the gray literature search that were not able to be uploaded to the software were digitized and analyzed independently by the 2 reviewers.

### 2.4. Data Processing and Analysis

The content extracted from the final bibliographic sample was entered into a Microsoft Excel table designed by the researchers and shared via cloud storage. It contained the following information: article title, author name(s), publication year, article type, objectives, methods, and main results/conclusions. 

After extraction of results, a thematic synthesis was carried out to organize the interventions according to their nature. 

A critical appraisal of the articles was not performed. The method does not oblige this evaluation. One of the peculiarities of this methodology is that it does not aim to analyze the methodological quality of the studies included, given that its objective, following the aforementioned, is not to find the best scientific evidence but rather to map the existing scientific evidence [11].

## 3. Results

In the first phase of the study, 220 articles were identified, of which 6 were duplicates. Reading titles and abstracts allowed the researchers to select 29 articles for full text reading. Application of the eligibility criteria to the articles available in the databases resulted in a set of six publications. Thirty-one documents were identified in the gray literature, of which two were included in the final sample (Figure 1). 

Table 3 shows information about the eight documents selected in the present scoping review, such as objectives, document type, and results that answer the research question. The bibliographic sample was heterogeneous and included five literature reviews [9,14,15,16,17], a survey [18], a qualitative study [19], and a book [20]. Two documents were produced in Portugal [9,20], one in Spain [14], one in Germany [19], one in the United States [18], one in the England [15], one in China [16], and one in Brazil [17].

Of the studies identified, two refer to interventions carried out in hospital [14,19], three refer to interventions carried out at home [15,16,18], and three contain interventions covering the transactional period [9,17,20].

Analysis of the contents of the selected articles identified interventions related to masks or interfaces [13,14,15,17], prevention of complications associated with NIV [13,15,16,18,19,20], leakage control [15,20], maintenance and cleaning of ventilators and accessories [13,20], respiratory training [20], ventilator monitoring [13,16], and communication and behavioral strategies [17,18,19,20] (Table 4).

## 4. Discussion

The present scoping review gathered articles and gray literature material to answer the research question. Although research on NIV use has been fruitful over past years, the sample indicated that the studies about the topic were biomedically oriented and predominantly carried out in the hospital setting. Transitional care programs focused on training clients and families in the transition from hospital to home in order to guarantee adherence of the former to NIV were not identified. 

Analysis of the articles showed mentions of interventions related to providing information to, training, and monitoring people so they could use and adapt to masks or interfaces [9,14,15,17], prevention of complications associated with NIV [9,15,16,18,19,20], leakage control [15,20], maintenance and cleaning of ventilators and accessories [9,20], respiratory training [16], ventilator monitoring [9,16], and communication and behavioral strategies [17,18,19,20] to promote continuous adherence to this therapy.

Transitional care programs centered on people who need to use NIV were not identified, which may contribute to the existing difficulties experienced by people in the community in terms of adhering to this therapy [20]. Since health systems aim to improve the quality of care delivered to clients, and consequently their experiences and clinical results, in order to minimize costs, it is urgent that care transitions between hospital and community be optimized for these patients. 

Preparation, which must be initiated in the hospital, must include health education sessions, follow-up appointments, phone calls, and support for home care [10,20,21] with individualized care plans to fulfill clients’ specific needs in order to minimize the chances of hospital readmission caused by worsening symptoms [15,16,21].

The discomfort and pressure caused by masks and the development of pressure injuries reinforce that crucial interventions should be part of the care plan, including guidance on correct mask adjustment, interface type switching, the choice of proper mask type and size, hygiene, skin hydration, and leakage control [9,14,15,17,19,20,22,23,24,25]. It is important to stress that, in addition to the quality of information provided to clients and their relatives, other factors affect treatment success, such as attention to comfort, pain control, collaboration with caregivers, and the quality of the relationship between healthcare professionals and clients/families [26]. 

A qualitative research study which interviewed 16 patients with chronic obstructive pulmonary disease (COPD) treated with NIV while hospitalized concluded that were some factors which promoted NIV tolerance, such as trust in the providers; a favorable impression of the facility and staff; understanding of why the mask was needed, how NIV works, and how long it will be needed; immediate relief of the threatening suffocating sensation; familiarity with similar treatments; use of meditation and mindfulness; and the realization that treatment was useful [27]. 

Studies also note that professionals need training and education concerning health issues related to the use of NIV [28,29] and how to educate and capacitate patients and their caregivers for the transition from hospital to community [10,12]; patients are more adherent with an intervention when they are involved in the decision-making process [30]. For these difficulties to be overcome, the authors proposed that the care plan be flexible enough to meet clients’ individual needs and that there be a leader in the process who can train professionals so they can deliver safe care and ensure that multiprofessional teams have effective introduction, training, and supervision [19,20], for example, by discussing clinical cases [19], supporting family-centered care training, and making sure clients are treated with empathy and respect [19,20].

Empowering the patient with information and explanations regarding physiologic need and tangible benefits prior to initiating the device can establish a trusting relationship and reduce doubts and fears about this therapy [27,30], increasing adherence to the use of masks and interfaces and preventing complications associated with long-term maintenance [9,15,16,18,19,20,27,30]. Comfort of the intervention is dependent upon how well the patient and device work together for optimal physiologic outcomes [30]. 

Our results shows that leakage due to poor adjustment of the mask or inadequate fit with the patient’s characteristics is a cause not only of nonadherence, but also of irritation and eye infections [15,20], which highlights the need for follow-ups with these patients at home for early detection of this problem and its complications. The discharge and transitional care plan includes a visit by the nurse in the first 2/3 days after discharge to assess the patient and the caregiver, their doubts and difficulties, and conditions at home [10], as well as precociously identify situations that pose a risk to adherence. 

A consensus statement from the Agency for Clinical Innovation, that will be reviewed in 2024, observes that patients and caregivers should have a minimum level of skills and knowledge during acclimatization to NIV [31], namely about the maintenance and cleaning of ventilators and the changing or cleaning of filters and accessories such as humidifiers and nebulizers. 

It should be noted that there are few studies on the follow-up of these patients at home and on interventions that can increase adherence to treatment and empower the patient to be independent in carrying out daily life activities. Future studies should explore this issue.

Another result from our SR concerns respiratory training with the aim of reducing dyspnea and consequent work of breathing, maintaining or improving tolerance to physical exercise, mobilizing secretions and assisting in their removal, and improving ventilation efficiency [20,30,31]. It is also important that these patients undergo muscle strengthening exercise training and activities of daily living training carried out with an energy compensation technique [30].

An aspect that did not emerge in the results, but that deserves attention in these patients who present a greater risk of falling [32], is the prevention of accidents. The transition plan and follow-up must provide for the assessment of risk factors, including environmental risk, advice on modifications, adaptation to space and equipment, and cognitive behavioral strategies that increase safety [10,32].

The results of the present scoping review allowed for verification that the main focus for clients in adherence to NIV should be on health education [9,11,14,15,16,18,19]. Given that nonadherence of clients to this practice results from lack of information, it is fundamental that nurses have sufficient scientific knowledge to formulate care plans that meet clients’ needs regarding the use of NIV. 

The aspects related to the relationship with the health team, communication, and behavioral strategies [17,18,19,20] indicated the need to speak in a soft and calm tone of voice and maintain visual contact and a calm attitude, which conveys self-confidence and competence, leading to clients feeling motivated to adhere to the treatment [25,33]. 

A study that had the objective of analyzing the impact of a brief psychological support intervention on adherence to NIV among patients with COPD concluded that the intervention group, which received cognitive behavioral therapy support, including counselling, relaxation, and mindfulness-based exercises, showed improvements regarding both adherence to NIV (F(304) = 19.054, *p* < 0.001) and quality of life (F(156) = 10.264, *p* = 0.002) after eight meetings compared to the control group. The results also showed a significant change in quality of life over time (F(71.480) = 8.114, *p* = 0.006) [33].

Future studies must explore the use of these strategies in improving adherence and training for clients and families so that they can be independent regarding NIV management, leading to improved self-care and execution of activities of daily living. Since the prevalence of people with NIV is high, and because it is estimated that prevalence will continue to increase as a result of the increasing prevalence of chronic respiratory diseases [9,19,20], as well as there being several difficulties with the use of masks and equipment and barriers to therapy continuity [5,7,8,14,15], it is recommended that transitional care programs to train clients and caregivers in NIV use be designed, implemented, and evaluated. 

Resorting to e-health programs can improve communication between clients and professionals [1] and help clarify questions in a timely manner [34].

This study has limitations. The restriction on language and free access to full texts may have excluded articles that answered the research question and respected the eligibility criteria, allowing some articles to be excluded deductively, thus missing important results that may have contributed to answering the research question and formulating recommendations. Another limitation is the option of not evaluating the methodological quality of the included articles, which is accepted in the method but limits the evidence of the results and the recommendations for practice.

Despite these limitations, the results allow the authors to identify the existing nursing interventions aimed at training clients and families to correctly use NIV in the transition from hospital to community and also allow for the proposal of recommendations for future studies that should address the difficulties experienced by people and their families in transitioning from the hospital to the community. Given the little evidence available on interventions in this period, qualitative studies are recommended [35] to bring attention to the voices of these people and their families to help identify strategies to focus care and allow adherence to NIV.

## 5. Conclusions

Hospital discharge planning must be initiated during hospital stay and aim to identify clients’ and family caregivers’ needs so they are met at home. For this goal to be achieved, it is necessary to implement several nursing interventions related to health education, specifically to necessary precautions concerning mask adjustment, equipment handling, prevention or resolution of complications that may emerge during the process, respiratory training, and cognitive behavioral strategies oriented toward reducing anxiety and clarifying questions, which are common issues for most NIV users. 

As a recommendation for clinical practice, the authors suggest continuing education of healthcare professionals. It is also important and necessary to inform multiprofessional teams about the presence of schematic and visual supporting materials intended to improve dissemination of practical information for the final objective of improving quality of care delivered to patients submitted to NIV, thus promoting greater adherence to using this type of therapy at home. 

Additionally, the present review points out the need to design new studies on transitional care programs that impact correct use of NIV at home. 

## Figures and Tables

**Figure 1 healthcare-12-00545-f001:**
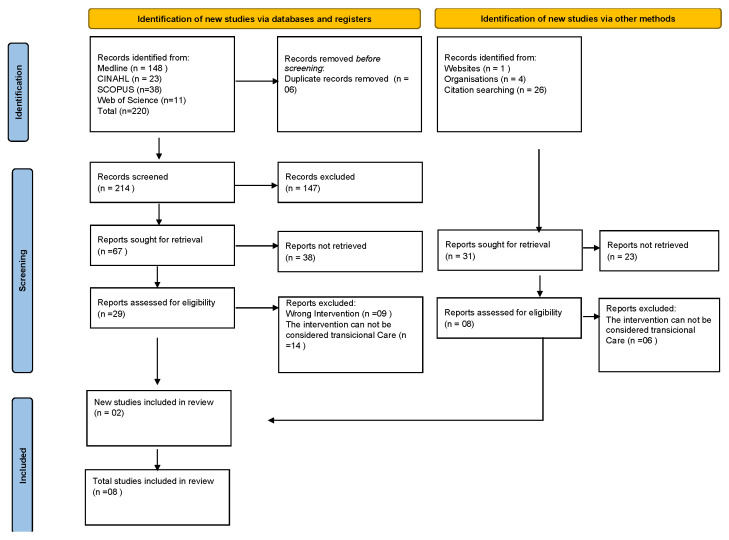
PRISMA-SR flowchart [13] (Lisbon, 2023).

**Table 1 healthcare-12-00545-t001:** Eligibility criteria, Lisbon, 2023.

	Inclusion Criterion	Exclusion Criterion
P	Adults and elderly people submitted to NIV.	Age < 18 years.
C	Interventions to qualify clients and family caregivers to use and manage NIV: education and training, monitoring, communication, and behavioral strategies.	Interventions to qualify clients and families to use and manage IV. Formal caregivers.
C	Transitional care between hospital and community (hospital stay, hospital discharge, transition from hospital to home, first 30 days after returning home).	Residential facilities for elderly people, long-term care in health or social institutions, and rehabilitation units.

**Table 2 healthcare-12-00545-t002:** Search strategy used in the MEDLINE database search (Lisbon, 2022).

	Search Strategy	Number of Articles
#1	(((((((((((((((((((Elderl*(Title/Abstract))) OR (aged(Title/Abstract))) OR (age*(Title/Abstract))) OR (older person*(Title/Abstract))) OR (older adult*(Title/Abstract))) OR (middle age(Title/Abstract))) OR (younger adult(Title/Abstract))) OR (frail older adults(Title/Abstract))) OR (aged(MeSH Terms))) OR (frail elderly(MeSH Terms))) OR (adult, frail older(MeSH Terms))) OR (adults, frail older(MeSH Terms))) OR (adult, young(MeSH Terms))) OR (adults, young(MeSH Terms))) OR (middle age(MeSH Terms))) OR (middle aged(MeSH Terms))) NOT (children(MeSH Terms))) NOT (adolescent(MeSH Terms))) NOT (adolescence(MeSH Terms)))) Filters: Free full text, in the last 5 years	15,148
#2	((((((((((((((((((((Mechanical ventilator(Title/Abstract)) OR (Cpap ventilation Positive end expiratory pressures(Title/Abstract))) OR (Mechanical ventilation(Title/Abstract))) OR (Biphasic continuous positive airway pressure(Title/Abstract))) OR (Bilevel continuous positive airway pressure(Title/Abstract))) OR (Inspiratory positive pressure ventilation(Title/Abstract))) OR (Positive end expiratory pressure(Title/Abstract))) OR (Non-invasive ventilation(Title/Abstract))) OR (Noninvasive ventilation(Title/Abstract))) OR (NIV(Title/Abstract))) OR (BIPAP(Title/Abstract))) OR (CPAP(Title/Abstract))) OR (IPAP(Title/Abstract))) OR (EPAP(Title/Abstract))) OR (PEEP(Title/Abstract))) OR (Biphasic positive airway pressure(Title/Abstract))) OR (Continuous positive airway pressure(Title/Abstract))) OR (Non-invasive positive pressure ventilation(Title/Abstract))) OR (Artificial Ventilation(Title/Abstract))) OR (non invasive positive pressure ventilation(MeSH Terms))) OR (positive pressure non invasive ventilation(MeSH Terms)) Filters: Free full text, in the last 5 years	20,668
#3	(((((((transitional care(Title/Abstract)) OR (discharge(Title/Abstract))) OR (patient discharge(Title/Abstract))) OR (TCM(Title/Abstract))) OR (home nursing(Title/Abstract))) OR (discharge planning(MeSH Terms))) OR (discharge, patient(MeSH Terms))) OR (aides, home care(MeSH Terms)) Filters: Free full text, in the last 5 years	47,391
#4	((((((intervention(Title/Abstract)) OR (capacitation(Title/Abstract))) OR (health education(Title/Abstract))) OR (information(Title/Abstract))) OR (early intervention education(MeSH Terms))) OR (habilitation(MeSH Terms))) OR (health education(MeSH Terms))Filters: Free full text, in the last 5 years	495,644
#5	#1 AND #2 AND #3 AND #4	148

**Table 3 healthcare-12-00545-t003:** Studies included in the scoping review (Lisbon, 2023).

Reference, Origin Country, and Publication Year	Study Design and Objectives	Results
[9] Portugal 2017	Integrative review	It is essential to define criteria for the care of people under NIV, including those for its prescription, maintenance, and evaluation. Equally fundamental are interface selection, initial setup, adjustments, and knowledge of NIV failure predictors. Noninvasive ventilation implies specific surveillance by clients, in which nurses play a prominent role.
[14] Spain 2017	Literature review. Identifying risk factors for incidence of skin injuries associated with clinical devices caused by NIV, preventive strategies to reduce them, and the most effective treatment for injuries that cannot be avoided.	The mask of choice was the facial one, and foam or hydrocolloid dressings were always applied on the nasal bridge. The condition of the skin under the interface and harness must be evaluated every 4 h (recommended) or every 11 h (maximum). The interface rotation strategy must be evaluated at 24 h if NIV is still needed on an ongoing basis.
[15] UK 2019	Systematic literature review. Evaluating the reason why clients needed long-term noninvasive positive pressure ventilation, describing some necessary nursing care procedures, and identifying some challenges experienced by nurses who provided home support when they interacted with these clients.	Noninvasive ventilation is a therapy that has been widely used in the home environment. Consequently, it is important that healthcare professionals understand the principles of its use in this setting and the challenges it can present for clients. Some crucial aspects that must be considered include the importance of proper mask adjustment, complications that clients can experience because of ventilation therapy, and how they can influence its effectiveness. Therefore, nurses are the professionals responsible for improving the way client care is managed at home by developing knowledge and specialized understanding of use of NIV in the domestic setting. This will promote client adherence, prevent complications, and decrease the number of hospital admissions. Using humidification units coupled to the ventilator can help some clients, but these units require daily cleaning. The container must be filled with boiled water so risk of contamination is reduced.
[18] United States 2019	Survey study. Collecting data about the experiences and care transition of clients, as well as factors associated with post-discharge follow-up.	There were inconsistencies in care transition processes. The authors recommended health education sessions, follow-up appointments, phone calls, and support for home care.
[19] Germany 2019	Qualitative study. Describing the quality of nursing care for clients submitted to NIV at home in Bavaria, Germany, and providing improvement recommendations from the perspective of health professionals.	This study described a heterogeneous and partly deficient care situation of people with NIV but showed that high quality care is possible if person-centered care is successfully implemented in all areas of service provision. Delivering person-centered care should be based on empowering ventilated patients to be completely involved in all decisions regarding their care and support. Care should support autonomy, focus on individuals’ needs and preferences, and enable ventilated patients and their families to consider treatment options and make informed decisions. Successful person-centered care initially requires the appropriate attitude to meet people’s needs (*outcomes and impact*), the involvement of ventilated patients in all decisions related to their care (*service delivery*), and a common vision of person-centered care provided by inspiring leadership (*vision and leadership*).
[16] China 2022	Literature review. Providing an overview of titration and follow-up of clients under noninvasive positive pressure ventilation. Focused on different technologies, modalities, managements, and cost-effectiveness used in Internet of Things-based telemonitoring of home mechanical ventilation.	Actively monitoring and communicating information during follow-up were crucial for long-term adherence. The medical Internet of Things will shift care from hospitals and clinics to homes and mobile devices. Patients may communicate with doctors or nurses at home via smart phones, mobile applications (apps), or the Internet. The Internet of Things allows users to access these “things” wherever and whenever they require them. There are a lot of opportunities for the Internet of Things to help remote caregivers ensure the safety of patients with noninvasive positive pressure ventilation and other wearable devices and raise warnings over critical situations. In this situation, providers should respond immediately to patient needs.
[17] Brazil 2022	Integrative review. Identifying the needs of critically ill patients who had to be submitted to NIV and their families, as well as nursing interventions that promoted adaptation of these patients and their families to NIV.	Adaptation of clients under NIV was promoted by implementing nursing interventions, both pharmacological and nonpharmacological. It is important to first adopt a nonpharmacological intervention, which includes four essential domains: communication, technology, comfort promotion, and environmental management. It is critical that nurses develop nursing technology competencies so that these professionals can allow technology and care to coexist harmoniously and develop activities oriented toward promoting clients’ physical integrity.
[20] Portugal 2021	Book with evidence on COPD. Identifying nursing care that must be implemented to help clients with COPD regarding use of NIV during exacerbations and at home and informing them about the importance of motor and cardiorespiratory training for clients’ activities of daily living with the purpose of maintaining their quality of life.	It is important to take a few breaks in the use of the mask for clients to moisten their face, humidify the oral mucosa, and clean the mouth with mouthwash. Some breathing techniques stood out, such as inhaling through the nose with the mouth closed, directing the air into the abdomen, carrying out diaphragmatic breathing, and slowly exhaling through the mouth with the lips half-closed. The study presented care procedures for different complications associated with NIV.

**Table 4 healthcare-12-00545-t004:** Systematization of nursing interventions intended to train clients and family caregivers in correctly using NIV in the transition from hospital to community (Lisbon, 2023).

Nursing Interventions to Train Clients and Families to Correctly Use NIV in the Transition from Hospital to Community
Masks or interfaces [9,14,15,17,20] Teaching of mask or interface placement and training in their adjustment. - Full face masks: upper edge supported on the nose wings, lower edge resting on the chin. - Nose mask: resting on the nose wings between the nose and the upper lip. - First adjust the mask and then the harness, without exerting too much pressure.
Prevention of complications associated with NIV [9,14,15,16,17,20] Pressure injuries: - Correctly moisturize the skin. - Check the skin condition every two hours. - Use hydrocolloid or polyurethane dressings in the areas under highest pressure. - Remove the mask intermittently to provide the areas under highest pressure with some relief. Eye discomfort: - Apply artificial tears and wet eye dressings. Nasal congestion: - Wash the nose with saline solution. - Apply steroids and antihistamines. Dry mouth and nasal mucosae: - Clean the mouth with mouthwash. - Orally hydrate the mucosae by drinking liquids. - Use humidification units coupled to the ventilator (use boiled water to reduce risk of contamination). Abdominal distension: - Keep the mouth closed when using the NIV device. - Promote mobilization and elimination of secretions. - Use antiflatulents if necessary. - Do exercises that recruit the lower limbs. Risk of vomit aspiration: - Watch the clients for some time after they take solid and liquid food. - Keep clients in Fowler’s position for at least 30 min after meals. - Remove the equipment in case of nausea or vomiting.
Leakage control [9,14,15,17,20] - Choose the mask correctly, opting for one with smaller dimensions, or change the mask type according to new needs and meet the requirements of each model. - During mask placement, make sure that it does not collide with the corners of the eyes and/or the mouth. - Observe the correct body position and make sure that the mask is placed when clients are in the proper position for receiving NIV. - Look around the mask in search of small leakages. - In case of the use of dental prosthesis, keep it in during the NIV procedure. - Avoid the presence of facial hair, namely beards. - Use fixation systems (for example, support for the chin).
Maintenance and cleaning of ventilators and accessories [17,20] - Clean the external surface of the ventilator with a wet cloth. - Wash the filters once a month with warm water and soap, then dry them properly before inserting them again. It is essential to replace them every six months. - Disassemble the mask and accessories fully once a week, wash whatever possible with warm water and soap, and dry it properly before assembling it again. - Clean the circuit externally with a wet cloth, without using cleaning products. - Place the equipment on a flat and stable surface in an airy, low-humidity place. - Always have a spare mask and circuit.
Respiratory training [9,20] Instructing in and monitoring breathing techniques: - Inhale through the nose with the mouth closed, directing the air into the abdomen. - Carry out diaphragmatic breathing. - Slowly exhale through the mouth with the lips half-closed. Positions for relaxation and respiratory control: - Sitting: feet firmly touching the floor, body slightly tilted forward, and elbows resting on the thighs. - Lying down: right lateral decubitus with the headboard elevated. - Standing up: elbows resting on a surface (for instance a low wall, a counter), with the body slightly tilted forward. -Semi-Fowler’s position. Techniques to clean the airway: - Inhale deeply. - Ensure proper hydration to guarantee secretion fluidification. - Carry out respiratory physical therapy. - Cough vigorously in one go, with the mouth open, to eliminate secretions.
Ventilator monitoring [9,14,15,16,19,20] - Reevaluate parameters up to three months after the beginning of the therapy. Subsequently, evaluation must occur annually. - Make sure the company that provided the equipment pays home visits at the beginning of the therapy and four weeks later. - Provide information about the presence of an alarm in the equipment, which allows identification of air leakage and consequent need to adjust the mask.
Communication and behavioral strategies [17,18,19,20] - Ensure a calm and quiet environment. - Speak in a soft and calm tone of voice. - Maintain eye contact with clients. - Keep a calm attitude, which conveys self-confidence and competence. - Promote resting and relaxation positions (for example, diaphragmatic breathing). - Learn how to put the mask on and attach it and do these procedures by yourself (if you are a client). - Provide information about the possible presence of alarms. - Suggest that clients try music therapy.

## Data Availability

The original contributions presented in the study are included in the article, and further inquiries can be directed to the corresponding author.

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
