# Peer review of "Nursing Interventions for Client and Family Training in the Proper Use of Noninvasive Ventilation in the Transition from Hospital to Community: A Scoping Review"

_healthcare, 2024, doi:10.3390/healthcare12050545_

Round 1

Reviewer 1 Report

Comments and Suggestions for Authors

Dear authors,

The paper addresses and important issue for nursing. And its relevant make  studies like this.

However, there are many concerns with the paper.

There are concerns related to the Background and theoretical basis for the study, the focus of the study and methodology. There are concerns about the organization of the paper, such as reporting of the Results vs. Discussion of the study findings and within the context of relevant previous research.

The Authors are encouraged to consider the suggestions for revisions. It's an important subject that deserves our attention.

The study requires a more explicit framework in the introduction about the concept of transitioning care, the preparation of family caregiver to take care and about self-management itself. Some background about the differences of health systems and responses in the community to this issue of care is needed too.

In the Introduction section, the reason to focus on the role of the nurse in improving the knowledge needs of both the patient and the family caregiver and consequently in the interventions aimed education and health literacy in these areas should be more clearly articulated.

Line 41, 42 – “Furthermore, there are difficulties in the adherence to non-invasive ventilation (NIV) [3]” More references are needed to support this statement and this link between the deaths and self-management of the NIV.

Line 46 to 50 need more references to support this statement.

Line 93,94 applying the PCC (Population, Concept, Context) need a reference.

The research question presents in lines 95 and 96 need to be improved. What is the concept of nursing interventions implemented to train families. The conceptualization is the family as a unit of care, as a client or the family caregiver? It is very important to clarify.

Exclusion criteria in P need to be more accurate, toddlers, and so on… other references to children… and in Context long-term care at home? It was an exclusion? Long-term care needs to be clarify.

Inclusion criteria in Context why you stablish 30 days? And the maintenance of the VNI at long in time it is not important? The some in line 115. In concept, families or family caregivers? The some in the line 110.

Line 120 “transitional care being a recent concern in healthcare” needs references.

Line 139 why Rayyan® it was important? References?

A narrative synthesis it was proper in systematic reviews with qualitative studies. When we read the results, I think do you made a thematic synthesis.

Note: you write that the literature are scares in this field, but you found (results) 5 in 8 literature reviews!

Table 3 show us the included studies, maybe the table need more information, such as the context of care if it was in hospital or at home or in the transitioning phase. In the results the interventions were patient/client centered care and not explicit the family caregiver.

Think about put the reference number in the interventions on table 4, it was more transparent in terms of which study talk about this specification of intervention. It was not clear what client was the target the patient or family caregiver. It was nice differentiate the interventions.

It seems to me that the first four paragraphs of the discussion bring back results in the form of data integration and interpretation.

Conclusions are centred in client/patient and not families.

have a good day

Author Response

Dear review:

We attached the answer.

Reviewer 2 Report

Comments and Suggestions for Authors

Thank you very much for inviting me to review the article “Nursing interventions for client and family training in proper 2 use of noninvasive ventilation in the transition from hospital to 3 community: A scoping review.” In general, the article is well written. However, kindly find my comments.

1.      I noticed a lot of spelling, punctuation, and grammar errors. I suggest a professional editing service to help improve the manuscript’s quality.

2.      The reference no.1 is the main reference for the data. Looks like the authors have taken reference no.4 from https://firsnet.org/images/publications/FIRS_Master_09202021.pdf. But they quoted this link.

3.      I suggest looking at the WHO website for more accurate data.

4.       The most common cause of death in COPD was chronic lower respiratory disease. This is totally unclear.

5.      Referencing may need to be improved. For example, line 56 to 63 is one paragraph with one reference. Why did the authors include the same reference in each sentence of the paragraph?

6.      Again, uniform abbreviations are not followed. For example, after NIV, the authors mentioned noninvasive ventilation in the objectives area again.

7.      The word “The present review” would be preferred over the “Secondary study.”

8.      For me, the methodology looks like a systematic review. Also, the authors clearly narrowed the topic. Hence, I suggest reconsidering the title. “A scoping review is a type of literature review that aims to map out and summarize existing research literature on a broad topic area. Unlike systematic reviews, which focus on specific research questions and follow a strict protocol for study selection and data synthesis, scoping reviews are more exploratory and aim to provide a comprehensive overview of the available evidence.”

9.      If the chosen publication date is from 2017 to 2023, it is six years, or the authors need to be more specific in dates.

10.   Again, in methodology, the eligibility criteria are ill-defined. The authors did not mention the type of articles that were aimed to include. I can see that a lot of included articles are review articles. This is against the rationale statement given by the authors. Hence, I suggest including more original articles in the results.

11.   The discussion is fairly written. However, it needs to be modified according to the modified (revised) results table.

Comments on the Quality of English Language

Dear authors,

Professional English editing is required as I can see a lot of spelling, punctuation, and grammatical errors. 

Author Response

Dear review:

We attached the answer.

Reviewer 3 Report

Comments and Suggestions for Authors

This is a valuable research however it need revision to be more suitable to be published in this journal.

 1. L 121. It was written " A full text filter and publications in English or Portuguese were included. "

 But in L177 the author stated "Two documents 177 were produced in Portugal [12,18], one in Spain [13], one in Germany [14], one in the 178 United States [15], one in England [16], one in China [17], and one in Brazil [19]."

 Please reorder inclusion criteria and exclusion criteria. The author stated it in PCC mnemonic form. But extra information should be given in detail especially for exclusion criteria. And please check whether exclusion criteria in concept is right.

 2. 1. L119. This study limited searching date period for 5 years. Although the author stated the reason in the manuscript this is the major problem of this study.

 Six articals and 2 gray literature were included in the final sample. From 8 studies 5 were literature review or integrative review or systemic review. This means that the results of this study were suggested from other review papers.

 And one book was included in the final sample. The author need to clarify the searching criteria. Was this study limited to the full text paper? Or were books newspapers, magazine etc. included?

3. Please include more details of previous researches of  associated with NIV.

Author Response

Dear review:

We attached the answer.

Round 2

Reviewer 1 Report

Comments and Suggestions for Authors

Thank You for take time to make some changes in te manuscript. 

Congratulations.

Author Response

Thank you for your review.

Reviewer 2 Report

Comments and Suggestions for Authors

Dear authors,

Thanks for your efforts in revising the paper.

Kindly add a reference for "PRISMA-SR flowchart. Lisbon, 2023."

Does it differ from regular PRISMA, 2020?

Wish you all the best and congrats for the great work.

Comments on the Quality of English Language

Quality has improved. 

Author Response

Thank you once again for reading our article and for your suggestion.

Kindly add a reference for "PRISMA-SR flowchart. Lisbon, 2023."

We added this reference (Figure 1. PRISMA-SR flowchart [13]. Lisbon, 2023)

 Tricco, AC, Lillie, E, Zarin, W, O'Brien, KK, Colquhoun, H, Levac, D, Moher, D, Peters, MD, Horsley, T, Weeks, L, Hempel, S et al. PRISMA extension for scoping reviews (PRISMA-ScR): checklist and explanation. Ann Intern Med. 2018,169(7):467-473. doi:10.7326/M18-0850.

Does it differ from regular PRISMA, 2020?

This version of PRISMA was developed especially for the reporting of scoping reviews, allowing that the search in the gray sources and it screening could be reported in this flowchart.

Wish you all the best and congrats for the great work.

 Thank you!

Reviewer 3 Report

Comments and Suggestions for Authors

Thank you for your well prepared revised paper. I would like to give minor requestions.

1. In L71-78, The reference from the previous study was presented from 8-10. But reference 9 was one of the final searched literature. The reveiwer recommend to change the reference 9 .

2. In table4 the authors have inserted reference at the main titles. However, this is a literature review paper, therefore, please insert the reference number at the sub-title.

3. In review response, the reviewer answered "In the database search we limited to articles written in Portuguese and English (we put this as filter in the database), but we accepted articles from other countries.". Please describe this in the manuscript.

4. " The methos allows to search grey literature, after the search in scientific databases 2023. In this phase of search for gray literature we identified the book, that meet the inclusion criteria"

-> If so, please describe that the include books newspapers, magazine etc. were included as grey literature after the search.

Author Response

Thank you for your well prepared revised paper. I would like to give minor requestions.
1.    In L71-78, The reference from the previous study was presented from 8-10. But reference 9 was one of the final searched literature. The reveiwer recommend to change the reference 9 .
In the first version of the article, we didn't include any references to the articles in the bibliographic sample in the introduction. In the second version of the article, in response to a request from one of the reviewers, it was necessary to mobilize information from the introduction of that article to the introduction, hence the reference appeared in the introduction and then in the table of included articles.
2.    In table4 the authors have inserted reference at the main titles. However, this is a literature review paper, therefore, please insert the reference number at the sub-title.
The thematic analysis was based on the articles mentioned. The references are associated with the article that allowed the topic to be defined.
3.    In review response, the reviewer answered "In the database search we limited to articles written in Portuguese and English (we put this as filter in the database), but we accepted articles from other countries.". Please describe this in the manuscript.
We added it.
4. " The methos allows to search grey literature, after the search in scientific databases 2023. In this phase of search for gray literature we identified the book, that meet the inclusion criteria"
-> If so, please describe that the include books newspapers, magazine etc. were included as grey literature after the search.
We added it.